# The 2021 National Report on Sports Innovation for Montenegro: Content Analysis

Borko Katanic [1,2], Sanja Pekovic [3,4], Radenko M. Matic [4,5,*], Jovan Vukovic [5], Bojan Masanovic [1] and Stevo Popovic [1,4]

1   Faculty for Sport and Physical Education, University of Montenegro, 81400 Niksic, Montenegro; borkok@ucg.ac.me (B.K.); bojanma@ucg.ac.me (B.M.); stevop@ucg.ac.me (S.P.)
2   Sport Innovation Lab, Faculty for Sport and Physical Education, University of Montenegro, 81400 Niksic, Montenegro
3   Faculty of Tourism and Hotel Management, University of Montenegro, 85330 Kotor, Montenegro; psanja@ucg.ac.me
4   Western Balkan Sport Innovation Lab, 81000 Podgorica, Montenegro
5   Faculty of Sport and Physical Education, University of Novi Sad, 21000 Novi Sad, Serbia; jovanvukovic89@gmail.com
*   Correspondence: radenkomatic@uns.ac.rs; Tel.: +38121450188

**Abstract:** This research aims to content analyze the literature on innovation in the field of sports in Montenegro to provide a better basis for the establishment of a monitoring system. The research was conducted drawing on a pre-established methodology related to the assessment of the level of sports innovation based on 16 defined indicators. Grades were attributed to data found in available scientific articles that were published until 2021, as well as in secondary data sources (Google Scholar, PubMed, Scopus, and Web of Science), such as governmental and nongovernmental reports and online content on Google (N = 18) from the same period. The findings of content analysis indicate that all indicators averaged 2.25 (fair) on a six-point scale, while only one indicator was rated as excellent, five indicators were rated as good, four indicators were rated as fair, and lastly, five indicators were rated as poor. Based on the obtained results, a low level of innovation in the field of sports in Montenegro was determined, and this also applies to innovation indicators individually. These results can be used as an initial step in planning an appropriate strategy development at the national level, which will lead to the improvement of innovation activities and their implementation in the field of sports in Montenegro.

**Keywords:** sport; innovation; report; indicators; monitoring; Montenegro

## 1. Introduction

Contemporary society is paying more and more attention to innovations. Etymologically, the term "innovation" comes from the word "novus", which means new or alternative, and it is associated with "a new idea, method or device" and "the process of introducing something new" [1]. It can be noticed that already in the definition of innovation there are two main characteristics of innovation, i.e., the existence of a great idea for innovation and the method of its use and application. These characteristics could be divided into invention—the existence of an idea—and innovation—translating an idea into application and use [2]. Today, innovation has a broader meaning and relates to an analytical concept related to various academic fields, such as political science, sociology, organization, business, management, etc. [3,4]. In general, innovation, as a property of the organization implies acceptance and openness to change, with no or little resistance to change. Changes occur as a need for the future development of the organization or as a reaction to impulses from the business environment. Consequently, innovation has become a determinant of the necessary entrepreneurial activity, for survival, competitive advantage, and profitability in the market [5].

Innovations are the subject of study in many disciplines, so each scientific discipline has adapted the definition of innovation from its point of view. However, what is common to all definitions is that, when it comes to innovation, it is about adopting new ideas, strategies, and business principles. Their importance is emphasized by both practitioners and scholars, and they consider them an indispensable condition for the growth and development of the organization, as well as a mechanism for gaining a competitive advantage [6,7].

There is an expansion of innovations lately, where due to prompt technological progress and enormous competition, the implementation of innovations has become a crucial factor for doing business [8]. Nowadays, innovations are used in all areas of the economy and are the main factors of economic growth, and encouraging innovation is one of the aims of the 2030 Agenda for Sustainable Development (Article 9) [9]. Innovation and innovative strategies, continuous improvement, and application of knowledge bring many advantages, and their importance is reflected in the following: they stimulate economic growth and create profit, as an innovation growth of 1% brings an increase in income per citizen of about 0.05%, and they affect the increase of employment, etc. [10,11]. Due to the impact on economic development, innovation is becoming an important area of research [8].

The sports industry is "a global industry affecting many other sectors of the economy" [12] (p. 238) and is viewed as "a multidisciplinary field that includes various disciplines such as marketing, finance, legal aspects, governance, communication, organizational behavior and theory, sport for development, tourism, facility management, and event management" [13] (p. 602). In this regard, the sports sector represents one of the key factors in the economic development of the European Union (EU), as well as in national economies [14]. This is supported by the fact that the sports sector has a significant share in the total gross domestic product (from 1.76% to 3%) in the EU, and the percentage employed in the sports field is 3.5% of total employment in the EU [15].

Analogous to the fact that innovations are implemented in all areas of the economy and are an essential part of entrepreneurship [16], the success of sports organizations is associated with their innovation activities as well as their ability to adjust to rapid market changes [17]. Innovations in the sports industry occur in many ways, including sports organizations, sports teams, and sports players [12]. Despite the amount of innovation that occurs in the sports industry and the significance of innovation to sports actors, there is still a lack of research on this topic [12]. Due to significant changes in the market, even sports organizations need to adjust their business strategies to stay competitive in the market. However, according to Stewart and Smith [18], sports organizations are not interested to adopt innovations, unless they are focused on sports sciences, i.e., if it will contribute to the improvement and give additional quality to the team in the field. In the case of other adjustments, sports organizations may be considered conservative and more connected to the traditional way of doing than other organizations. Additionally, the variability of a sports product contributes to new differences concerning organizations in other industries, because sports organizations can find it somewhat more difficult to guarantee the quality of a sports product or service compared to other organizations that produce consumer goods or services [19]. In addition, in the field of sports matching, outcomes and their quality are often uncertain. At the same time, one competitor or club can dominate a competing club, which can lead to a decrease in the attractiveness of the match (especially for the fans of the athlete or the losing club). Hoye, Smith, Westerbeek, Stewart, and Nicholson [20] argue that the means of telecommunications and other means of technological development contribute to the need for sports managers to realize their potential, and readily accept the constant changes that result from globalization, politics, and their professionalization. According to Covell, Walker, Siciliano, and Hess [21], sports organizations need to be creative and innovative to differentiate themselves from the competitors to stay competitive. Improvement or diversity is provided by innovations as a necessary segment of development plans of sports organizations. The authors especially

emphasize the huge impact of advances in technology on sports, modernization of rules, and better public access to all necessary information.

Nowadays, innovation is often measured and can be viewed from several perspective—those of companies, economy, branch of activity, and regional and world aspects. Indicators are obtained based on which steps can be created for more successful business of the company, i.e., the region or the state [22]. On the other hand, according to Tidd [23], there are several types of barriers to innovation: (1) economic—personal costs, access to information, and insufficient incentives; (2) behavior—priorities, motivation, rationality, inertia, tendency to change, or risk; (3) organizational—goals, routine, power and influence, culture, and stakeholders; and (4) structural—infrastructure, non-refundable costs, and management.

Overall, there is currently no research assessing innovation in sports, although there is a need for it. There is no established system for assessing innovations in the field of sports in Montenegro and the wider region, so it is unclear what is the current level of innovation in the field of sports in Montenegro. Thus, the planned study could bring a new methodology for covering the existing gap in knowledge in the evaluation of the current status of sports innovations in Montenegro. At the same time, this study has an intention to better explanate the concept of sports innovations. Lastly, this study does not have the opportunity for comparison with some similar research from this country in the previous period, so this paper could be an initial contribution for comparison in the future.

*Economic Type, Global Entrepreneurship Index, and Hofstede Values*

Regarding the type of economy from The World Bank from 2021, Montenegro belongs to the group of upper-middle-income countries [24]. The Global Entrepreneurship Index Rank of All Countries [25] reveals a moderate relationship and support to entrepreneurship ecosystems and showed that this index in Montenegro took 57th place with 31.8 points.

Based on Hofstede's cultural dimensions theory in Figure 1, six scores provide a better understanding of the effects of society's culture on the values and behavior of inhabitants of Montenegro. Thus, the high result in Power Distance of 88, reveals that in Montenegro powerholders are very distant in society, which provides them with more privileges. The next score showed a low result in the Individualism of 24, emphasizing that collectivist culture exists, with strong relationships among members in the group and intensive loyalty. Further, the intermediate score in Masculinity of 48 shows that in Montenegro there is no dominant cultural value (masculine for certain and feminine for other parts). The fourth indicator of Uncertainty Avoidance with a score of 90 reveals that inhabitants have a low level of readiness to accept changes. Thus, it is evident that there are rigid codes of belief, which are intolerant to non-traditional behavior. The five indicators of Long-Term Orientation with a score of 75 emphasized the pragmatic culture and propensity to save and invest. The last indicator of Indulgence (score of 20) showed great restraint.

Based on the above, there was a need to collect all available data at the national level, using a certain methodology, and conduct a content analysis to determine the level of sports innovation in Montenegro, as well as to make recommendations for improving this area. Thus, this paper aims to analyze all relevant and available scientific sources to evaluate the current status of sports innovation, intending to set an initial point for future analyses of progress.

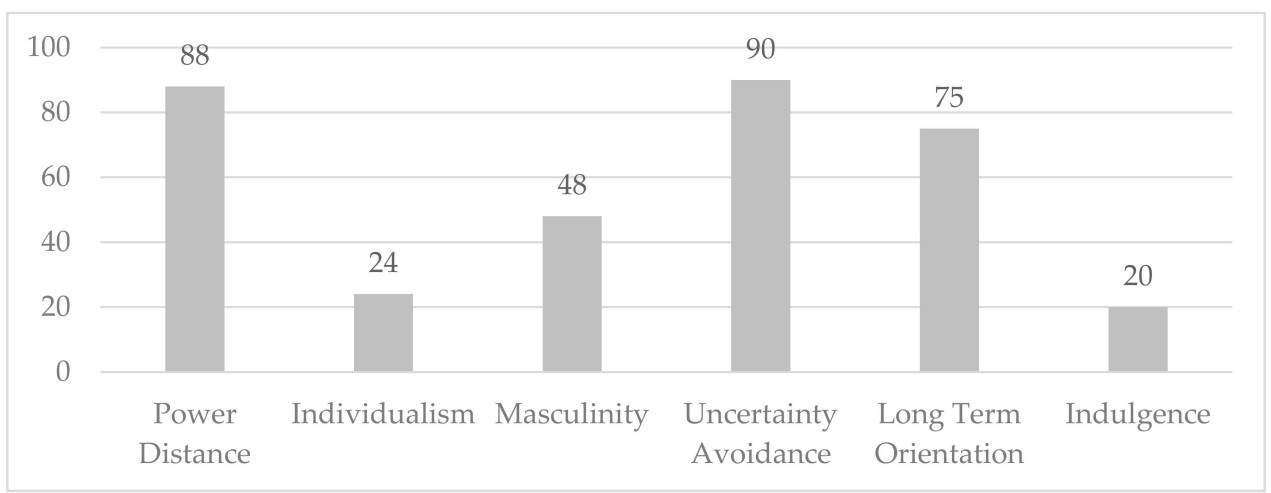

**Figure 1.** Hofstede values in Montenegro.

## 2. Materials and Methods

The research team established within the national project under the title "Montenegrin Platform for Innovation in Sports" (approved by the Ministry of Science in Montenegro: No.03/1-062/20-263/2 from 28 April 2020), set up a new methodology for assessing innovation in sports. The methodology was developed according to the leading methodologies for the assessment of general innovations in the countries of Europe and the world [26–28] and is described in detail in a separate document [29]. In this methodology, 16 indicators were singled out for the assessment of innovations in sports: innovative service/product, innovative working methods, organizational and marketing innovations, financial support for innovations, spending annually on innovation, income from innovation, higher education, new Ph.D. graduates, cooperation with universities, the person responsible for innovations, international scientific co-publications, international inventions, research and development (R&D) as a source of knowledge, innovation reports for the country, and scientific papers on innovations in sport. To evaluate each of these indicators, a separate content analysis was conducted, and a six-point grading scale was employed (5 = excellent; 4 = very good; 3 = good; 2 = fair; 1 = poor; 0 = without reliable information), as described in the previously mentioned paper [29]. The grades were awarded based on data found in available scientific articles in the last 10 years as well as in the secondary data sources such as governmental and nongovernmental reports and online content from a specific period. Reliability of content analysis included all three suggestions: (1) operationalization of the concept of innovation in sports in the protocol of the study, (2) training coders for implementation of the formulated concept of sports innovation, and (3) evaluation of the implementation over the reliability of coders [30]. The potential problem of risks of bias was solved by calculating the inter-judge agreement index using the Cohen's kappa coefficient ($\kappa$) suggested by McHugh [31]. Recommended values of this coefficient should be between 0.81 and 1.00 as almost perfect agreement. The electronic databases (Google Scholar, PubMed, Scopus, and Web of Science) were searched for research articles, and Google was searched for government and nongovernment reports and online content needed to evaluate the 16 indicators (Figure 2).

When a conclusion had to be drawn from multiple studies, the grade was a weighted average between the groups. The data were synthesized, and a set of benchmarks was used to assign the grades. Two researchers rated the studies, and if there were any deviations, the final grades were completed upon consensus.

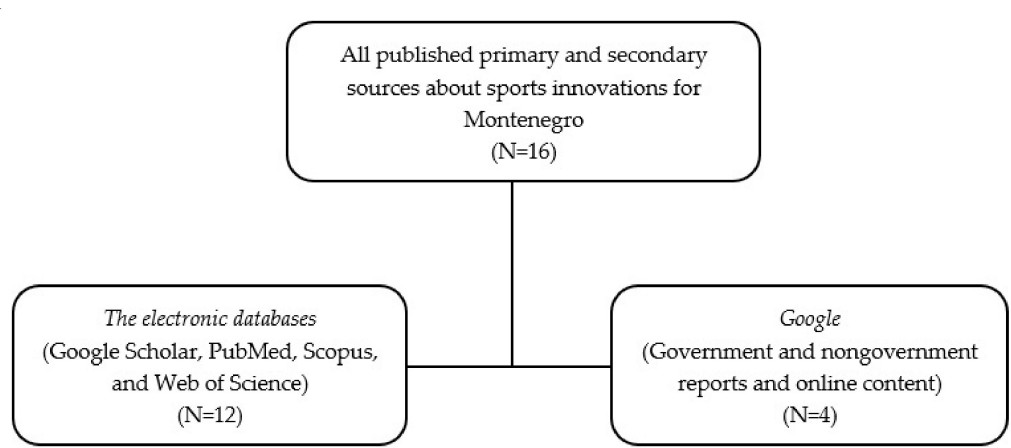

**Figure 2.** All published sources about all 16 indicators for sport innovations in Montenegro.

### 3. Results

At beginning of the interpretation of the data, all researchers compared their grades and determined that their initial inter-judge agreement Cohen's kappa coefficient (κ) was 0.91, which means almost perfect agreement. After a review of all source data, the researchers found additional information that contributed to maximizing the inter-judge agreement (k = 1). An accepted and big percent of intercoder reliability confirmed that all coders were well-introduced in the protocol of coding and well-introduced to codebook in this study. The results are presented in the form of grades for each indicator, which were awarded based on different numbers and the quality of selected documents (research articles, government, nongovernment reports, and online content). The ratings for each indicator are shown in Table 1.

**Table 1.** The National Report Grades.

| | Source data | Indicators | N | Tools | Results | The Innov. Service/ Product Indicator |
|---|---|---|---|---|---|---|
| 1. | Popovic, Bjelica, Zarubica, Pekovic, and Matic (2021) | Innovative service/product | 75 | Community Innovation Survey (CIS) | A total of 42.7% of respondents stated that, in the previous three years, the organization had introduced new or significantly improved products or services. | Grade 3 |
| 2. | Popovic, Bjelica, Zarubica, Pekovic, and Matic (2021) | Innovative working methods | 75 | Community Innovation Survey (CIS) | A total of 45.3% of sports organizations e introduced new or significantly improved working in the previous three years. | Grade 3 |
| 3. | Popovic, Bjelica, Zarubica, Pekovic, and Matic (2021) | Organizational innovations | 75 | Community Innovation Survey (CIS) | A total of 56% of respondents stated that, in the previous three years, the organization had new or significantly improved marketing and organizational activities. | Grade 3 |
| 4. | Popovic, Bjelica, Zarubica, Pekovic, and Matic (2021) | Marketing innovations | 75 | Community Innovation Survey (CIS) | A total of 50.7% of respondents stated that, in the previous three years, the organization had new or significantly improved marketing and organizational activities | Grade 3 |
| 5. | Popovic, Bjelica, Zarubica, Pekovic, and Matic (2021) | Financial support for innovations | 75 | Community Innovation Survey (CIS) | Only 21.3% of employees indicated that their organization received any form of financial support for innovative activities. | Grade 2 |
| 6. | Popovic, Bjelica, Zarubica, Pekovic, and Matic (2021) | Spend annually on innovation | 75 | Community Innovation Survey (CIS) | A total of 62.8% sport organizations spend less than 3% from the total profit annually on innovation. | Grade 1 |
| 7. | Popovic, Bjelica, Zarubica, Pekovic, and Matic (2021) | Income from innovation | 75 | Community Innovation Survey (CIS) | A total of 45.3% of respondents answered that there is no such income. | Grade 1 |
| 8. | Jovanovic (2012) | Higher education | 88 | Questionnaire | Only 37.5% of employees have an appropriate professional education. | Grade 2 |
| 9. | European Commission, (2020) | New Ph.D. graduates | - | - | According to official data from European Commission report for Montenegro in 2019, there are 0.0 new doctorate graduates per 1000 population aged 25–34. | Grade 1 |

**Table 1.** *Cont.*

| | Source data | Indicators | N | Tools | Results | The Innov. Service/ Product Indicator |
|---|---|---|---|---|---|---|
| 10. | Popovic, Bjelica, Zarubica, Pekovic, and Matic (2021) | Cooperate with universities | 75 | Community Innovation Survey (CIS) | Only 26.7% of respondents from sports organizations stated that they had established cooperation with universities. | Grade 2 |
| 11. | Popovic, Bjelica, Zarubica, Pekovic, and Matic (2021) | A person responsible for innovations | 75 | Community Innovation Survey (CIS) | Overall, 41.3% person(s) employed in sports organization are responsible for innovative activities. | Grade 3 |
| 12. | European Commission, (2020) | International scientific co-publications | - | - | According to official data from European Commission report for Montenegro in 2019, there are 70.9 scientific co-publications per million population. | Grade 1 |
| 13. | European Commission, (2020) | International inventions | - | - | According to official data from European Commission report for Montenegro in 2019, there are 31.9 registered patent applications per million inhabitants. | Grade 2 |
| 14. | Popovic, Bjelica, Zarubica, Pekovic, and Matic (2021) | R&D as a source of knowledge | 75 | Community Innovation Survey (CIS) | A total of 20% of respondents recognize internal sources as the most important source of knowledge that leads to the development of innovation in the organization. | Grade 1 |
| 15. | WIPO, (2020), World Economic Forum, (2020), and European Commission, 2020). | Innovation reports for country | - | - | We found five different global innovation reports, and 3 out of 5 reports have data for Montenegro (60%). | Grade 3 |
| 16. | Matic, Popovic, Pekovic, and Milovanovic, (2021); Popovic, Bjelica, Pekovic, and Matic, (2021); Popovic, Bjelica, Zarubica, Pekovic, and Matic (2021) | Scientific papers on innovations in sport | 75 | - Community Innovation Survey (CIS) | We found 91 papers on innovations in sport, of which 3 are by domestic authors. That is, the share of domestic papers represents 3.3% of all papers on a given topic. | Grade 5 |
| Average result | The overall average score for sports innovation for Montenegro is 2.2. | | | | | |

Note. Grade 5 = excellent; 4 = very good; 3 = good; 2 = fair; 1 = poor; 0 = without reliable information.

*Innovative service/product*

In a study published by Popovic et al. [14], objective tools were used to evaluate the percentage of a sports organization that created a new product/service in the last three years in Montenegro. Seventy-five subjects representing sports organizations were randomly distributed as a sample participated. The subjects came from all geographical parts of Montenegro and "represented their sports organizations as executive directors, presidents, secretaries, founders, and similar who have a basic knowledge of the main business flows of their organization and potential innovative activities" [14] (p. 96). The results showed that 42.7% of respondents stated that, in the previous three years, the organization introduced new or significantly improved products or services. The innovative service/product indicator was scored as good (grade 3).

*Innovative working methods*

Based on the study mentioned in the previous paragraph [14], an evaluation of the innovative working methods in sport organizations was performed. The results show that 45.3% of respondents stated that, in the previous three years, the organization introduced new or significantly improved working methods. Additionally, this indicator was in the range of 41% to 60% and was also scored as good (grade 3).

*Organizational innovations*

Organizational innovations in sports in the broadest context included several types of innovations (technical—technological aspects, or management, marketing, organizational, and cultural aspects of innovations, etc.) based on the study [14]. Since there were no other available official or scientific documents to describe this indicator, based on the results of the only one available study, it was found that 56% of respondents claimed that, in the previous three years, the organization had new or significantly improved organizational activities. Otherwise, 44% of respondents claimed that the organization did not introduce any innovations in organizational methods. Obtained results revealed that organizational innovation is rated with a good score (grade 3).

*Marketing innovations*

In the study published by Popovic et al. [14], marketing innovation in sports organizations in Montenegro was scored as good (grade 3). The results of a pioneer study in the area of sports innovations in Montenegro and only one available document that described this indicator showed that 50.7% of respondents stated that, in the previous three years, the organization had new or significantly improved marketing activities. This means that it ranged between 41% and 60%.

*Financial support for innovations*

For this important indicator, data for Montenegro were also found in the study of Popovic et al. [14]. The results show that only 21.3% of respondents answered that they received any form of public financial support for innovative activities in the last three years, while the rest did not receive any financial support. Additionally, it should be noted that this indicator has some limitations, which imply whether the organization received financial support or not, but did not refer to the amount of financial support. This indicator was scored as fair (grade 2).

*Spend annually on innovation*

There is also just one document that describes this indicator and that is the research study of Popovic et al. [14]. Results of the study showed that 62.8% of sports organizations spend less than 3% of the total profit annually on innovation. This means that this indicator was scored as poor (grade 1). The 62.8% were distributed as follows: 16% to allocate 1%–3% of total profit, while 13.5% stated that they allocate less than 1%, and as many as 33.3% did not allocate at all. On the other hand, a small percent of respondents, such as 20%,

pointed out that they allocate 3%–5% of profit, and 17.3% of respondents answered that they allocate over 5% of the total profit.

*Income from innovation*

Although "income from innovation" is probably the most important indicator in the used methodology, there was also just one available document [14] that describes this indicator. Results of this study show that as many as 45.3% of respondents answered that there was no income from innovation (1). However, it should be noted that 7% of respondents answered that they earned over 50% of their total earnings from innovations. This percent of respondents is small, and it generally does not represent the global picture of the situation in Montenegro; however, it should not be overlooked that some sports organizations have a huge profit from innovative activities, and it can be a great example of how it should be done in the future with other organizations. However, this indicator was also scored as poor (grade 1).

*Higher education*

Results of the study conducted by Jovanovic [32] showed interesting facts that, out of 88 respondents, as many as 55 did not have adequate professional education, which is 62.50%. These data represent the fact that only 37.5% of employees in sport organizations had the appropriate professional education in the field of sport management or related fields. For this reason, this indicator was scored as fair (grade 2). It should be added that a smaller percentage of employees had higher education but from other disciplines. On the other hand, it is very important to highlight that the research was conducted on sports organizations located in the northern municipalities of Montenegro: Bijelo Polje, Mojkovac, and Kolasin. Therefore, this indicator does not cover different regions in Montenegro, and it can represent a serious limitation in reaching the objective results for Montenegro as a whole.

*New Ph.D. graduates*

This indicator also refers to human resources, and the only available data were looked up in the official document of the European Commission in 2019 for Montenegro [26]. The data show that Montenegro had no new Ph.D. graduates per 1000 population aged between 25 and 34 years old in 2019. These data also confirm that there were no new doctorate graduates in the area of sports innovation. For this reason, this indicator could not be scored more than poor (grade 1).

*Cooperate with universities*

This indicator evaluates the percentage of sports organizations that have established cooperation with universities. Based on the study of Popovic et al. [14], the results showed that only 26.7% of respondents from sports organizations stated that they had established cooperation with universities, which resulted in a fair score (grade 2) because it is in the range of 21% to 40%.

*A person responsible for innovations*

Based on the only one available document, the research study that employed a survey that used 75 representatives from sports organizations in Montenegro [14], it was found that 41.3% of sports organizations have an employee (one or more) responsible for innovative activities, which implies that 58.7% do not have an employee for innovative activities. This indicator is in the range of 41–60%, and it was scored as good (grade 3).

*International scientific co-publications*

This indicator was read from the official European Commission report for Montenegro in 2019 [26]. The data were originally calculated by Science-Metric as part of a contract with the European Commission (EC) based on the Scopus and Web of Science databases. In this official document we found that Montenegro had just 70.9 scientific co-publications per million population in 2019. Although no data were available relating exclusively to

co-publications in the field of sports innovation, the general value was taken, as it was assumed that no significant deviations will occur from the general value. According to the scaling of this value on a six-point scale, a poor score was given for this indicator (grade 1).

### International inventions

According to the same official data from the European Commission report for Montenegro in 2019 [26], which are based on data from the European Patent Office (EPO) and Eurostat [33], it was determined that there were 31.9 registered patent applications per million inhabitants for Montenegro in 2019. According to the scaling of this value on a six-point scale and the same approach as was applied for the previous indicator, a fair score (grade 2) was given for this indicator.

### R&D as a source of knowledge

Only 20% of respondents recognized research and development (R&D) as the most important source of knowledge that leads to the development of innovation in the organization. This fact is available in the research study [14] that represents the only document that was used in reviewing this indicator in Montenegro. Furthermore, it is interesting to highlight that 80% of respondents did not agree with this, or had different opinions. Thus, it is interesting to note that most of them (33.3% of respondents) recognized internal sources as the most important source of knowledge, and 32% of respondents believed that these are personal and informal contacts with other organizations and colleagues from the region, 1.3% believed that these are customers, while 13.3% of respondents did not respond specifically. From the results mentioned above, this indicator could not be scored more than poor (grade 1).

### Innovation reports for country

All leading European and global reports were reviewed, in which the level of innovation for the country was analyzed. It was noticed that Montenegro is found in three of the five leading world reports, namely the Global Innovation Index [27], Global Competitiveness Report [28], and European Innovation Scoreboard [26]. It means that Montenegro is included in 60% of the world's research on the level of innovation in countries. Based on the given percentage, this indicator is scored as good (grade 3). Additionally, it should be mentioned that in all of these global reports, Montenegro is a modest innovator (European Innovation Scoreboard), and the sports innovations were not reviewed.

### Scientific papers on innovations in sports

For this indicator, papers were searched in databases specified in the method section, and a total of 91 papers dealing with innovations in the field of sports were found. Three papers (out of 91) were domestic papers on a given topic [14,34,35]. It should be noted that all of these three studies were created as part of the national project under the title "Montenegrin Platform for Innovation in Sports" that was approved by the Ministry of Science in Montenegro (No.03/1-062/20-263/2 from 28 April 2020). However, the share of domestic papers represents 3.3% of all papers on the given topic. Concerning the large number of countries in the world, these percentages are rated with a high score for this indicator (grade 5). However, since apart from the mentioned project, there is no research by domestic authors from Montenegro on the given topic, this indicator should be taken doubtfully, as it is not certain that it will be sustainable.

### Average result

In this research study, 16 indicators were reviewed that indicate the state of innovation in sports in Montenegro. A total average score of these 16 indicators was calculated and amounted to 2.25, which in a way represents a fair level of innovation in sports for Montenegro. In the case of Montenegro, this overall average score was expectedly low because all but one of the indicators were rated with grades from 1 to 3. It should be noted that there were no unrated indicators, which is a very good sign for a pioneer study in this interdisciplinary field.

## 4. Discussion

Based on the obtained findings, we may contend that the overall score for the level of innovation in sports in Montenegro is fair (2.25). This result is in line with the assessments of the leading world lists that rank countries in terms of general innovation, which confirms that the situation in Montenegro is not favorable. Thus, according to the European Innovation Scoreboard, Montenegro is marked as a modest innovator and ranked among the weakest group of countries [26]. Additionally, according to the Global Competitiveness Report, Montenegro is ranked 73rd (out of 141 countries), and when it comes to the area of innovation capacity, Montenegro is ranked 69th on the list World Economic Forum [28]. The slightly better placement was achieved on the Global Innovation Index, where Montenegro is ranked 49th (out of 131 countries) in innovation output, and 53rd when it comes to innovation input on the list WIPO [27]. Therefore, all this indicates, regarding general innovation, that the situation in Montenegro is not very enviable and is one of the main reasons why the sports field cannot be better positioned when it comes to innovation in Montenegro. Therefore, the government must emphasize innovation, because only the improvement of the country's innovation will lead to the improvement of the innovation of the organizations themselves [22]. This is especially important when it is known that innovation and the capacity to implement innovations are decisive factors in achieving the top performance of an organization [36].

The situation is similar when analyzing indicators individually. Montenegro has shown an extremely poor result regarding new doctorate graduates, which is well below the average values in EU countries, which amounts to 1.9 Doctors of Science in 2019 per 1000 inhabitants [26]. However, it is worrying that related to this parameter, Montenegro is below all European countries, grouped with the weakest EU countries such as Lithuania, Macedonia, Turkey, and Malta that have 0.3 new doctorate graduates per 1000 inhabitants [33]. When it comes to international scientific co-publications, it should be noted that most EU countries have a better average score than Montenegro, but it should also be emphasized that the achieved values correspond to the results of countries in the region [26]. However, if other related parameters were added, it could be noted that the indicators such as "scientific publications" places Montenegro at the bottom of the list as the 134th country [26], and "citation of papers with H-index" at the 128th place in the world, which is marked as one of the weakest [27]. Additionally, unsatisfactory results were achieved in cooperation with universities—in this study, Montenegro was scored as fair. Likewise, the picture is not good enough when it comes to cooperation with universities in all areas of the economy in Montenegro, based on which the country is ranked at the 51st place in the world [27]. All this may speak in favor of insufficient quality work of the universities themselves, which corresponds to the data that, according to research institutions prominence, Montenegro is among the lowest-ranked countries, more precisely in 102nd place in the world [26]. Further, it can be considered that there is insufficient engagement of academic institutions in various economic activities of Montenegro, which would lead to the exchange and implementation of certain knowledge, which would encourage various innovative solutions in various fields. However, this certainly cannot be the only reason for poor results. It should be borne in mind that the gross expenditure on research and development as the percent of GDP for Montenegro is 0.4%, and the country is ranked 73rd in the world [26], which indicates that society in Montenegro still does not recognize innovation and that the climate for innovation is unfavorable.

The situation is no better in terms of other indicators, so for the international inventions, Montenegro is rated fair, and according to the world lists, Montenegro is ranked 52nd [26] and 67th, respectively [27], by the number of patent applications per million inhabitants. It should be noted that some indicators correspond to the average values of European and world countries. The situation is somewhat better when it comes to tertiary education, whereas in Montenegro, 37.5% of employees in the sports sector have higher education, which corresponds to the average values in the EU in the field of sports, where 39% of employees completed tertiary education [37]. As for innovation reports for the country,

although Montenegro received a solid score for appearing in leading documents, this figure should be taken with a grain of salt, because it was seen that in all these reports it was rated very poorly.

It was not possible to compare other indicators because there are no data. All innovations within sports organizations related to innovative service and product, working methods, and organizational and marketing innovations are scored as good (grade 3). Still, companies try to introduce various innovations as much as possible, which is very important. On the other hand, this should be taken with caution, because the innovative product does not necessarily have to be new, but it is enough to be significantly improved, so it is classified as an innovation. Hence, this method of assessment resulted in a higher level of innovation. At least, it is necessary to take as very important parameters that financial support for innovations was assessed as fair and that revenue from innovations was assessed as poor (grade 1). These two important factors indicate that the state still does not recognize innovation sufficiently, which is in line with the above information. On the other hand, revenues from innovations are very weak, and the reason for this should be sought perhaps in insufficiently well-defined indicators, i.e., that innovative products and services are not a matter of special innovation but certain variants of the same product. Thus, when the given problem is differentiated, i.e., when innovation is considered as a completely new product, it will be clear how much could be earned from the innovation itself.

Several indicators from this study were rated the weakest: "spend annually on innovation", "income from innovation", "new Ph.D. graduates", "international scientific co-publications", and "R&D as a source of knowledge"; these correspond to the results that indicate that Montenegro was the weakest in the areas of human capital and research, knowledge, and technology outputs, as well as business sophistication [27].

If it is taken into account that Montenegro belongs to the group of countries with the lowest GDP [26], as well as the aforementioned insufficient positions when it comes to general innovations, it is clear that the authors of this study did not expect good conditions, even when sports innovations are in question. Nevertheless, this report can provide clear directions for improving some aspects of sports innovations. The best indicator in this national report of sports innovations is represented in scientific papers, as five indicators were evaluated with grade 3, four indicators with grade 2, and five indicators with grade 1. Probably, some aspects need more time for development, but some aspects can be upgraded in this process a little bit faster. The overall average score for sports innovation is 2.2, closer to fair than good. Thus, these findings and further knowledge suggest that Montenegro should be striving to adopt sports innovations more quickly. According to the interdisciplinarity of this topic, future research should include more parameters (social, economic, political, demographical, etc.) for analysis.

## 5. Conclusions

Based on the results that determined the low level of innovation in the field of sports, the assumptions from the overall results, as well as for the indicators individually, were confirmed. From this reason, it should be noted that this study is of great importance not only for practitioners and policymakers, but also for researchers, mostly because it identified the current situation in Montenegro by reviewing all available documents such as scientific articles, secondary literature, and online content, as well as world-leading reports. The conclusions of this study could be used as an initial step in devising stimulative measures and planning a recovery strategy for development at the national level, which would lead to the improvement of innovations and their implementation in the field of sports in Montenegro. The main guidelines of the future strategy should be based on the fact that this study identified several conclusions that might set priorities for its improvement. First of all more attention should be paid to the fact that the government does not essentially recognize innovations in sports and does not allocate enough financial resources, as well as that there is no awareness from stakeholders that income from innovations in sports might

be of great benefit. However, it was confirmed that these incomes are extremely low in Montenegro and a large space is open to progress in this direction.

The limitation of this study is reflected in the small number of documents that were available for analysis. There were not too many scientific papers, but also not many governmental and nongovernmental reports or online content, which is especially surprising. Therefore, this study not only officially confirmed that research on sports innovation is neglected in Montenegro, but also unequivocally invites all interested parties from Montenegro to invest in the areas of human capital and research, knowledge and technology outputs, and business sophistication in near future.

This study is of great value because it combines all available data in the field of sports innovation in Montenegro and represents a serious national report. This study as a pioneer in this field can provide a basis for authors from other countries to explore the given field of sports innovations and make annual national reports in their countries according to the established methodology, as well as to set up an international professional network. It would certainly help to gain a better insight into this interdisciplinary field and would also be important for easier mutual comparison of levels of innovation in sports between various countries from year to year.

**Author Contributions:** Conceptualization, B.K. and S.P. (Stevo Popovic); methodology, B.K. and S.P. (Stevo Popovic); formal analysis, S.P. (Stevo Popovic), R.M.M. and B.M.; writing—original draft preparation, B.K. and J.V.; writing—review and editing, S.P. (Sanja Pekovic), S.P. (Stevo Popovic), R.M.M. and B.M.; project administration, J.V. and B.K.; funding acquisition, B.M. and B.K. All authors have read and agreed to the published version of the manuscript.

**Funding:** This research received no external funding.

**Institutional Review Board Statement:** Not applicable.

**Informed Consent Statement:** Not applicable.

**Data Availability Statement:** Data are available upon reasonable request from the corresponding authors.

**Acknowledgments:** This research was completed within a national project under the title "Montenegrin Platform for Innovation in Sport" that was approved by the Ministry of Science in Montenegro (No.03/1-062/20-263/2 from 28 April 2020), as well as in line with the SHIINE COST Action's objectives (CA18236).

**Conflicts of Interest:** The authors declare no conflict of interest.

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
