# Peer review of "The 2021 National Report on Sports Innovation for Montenegro: Content Analysis"

_sustainability, doi:10.3390/su14042463_

Round 1

Reviewer 1 Report

Here are some suggestions and proposals for improvements from the review of the manuscript.

Introduction: it would be useful to include the initial assumption for the work and its main objective in the introduction to the article.

Methodology: as the methodology for the article is based mainly on a review of the scientific literature and the selection of articles and documents related to the issue being studied, the article should be adapted to reflect the main guidance in the Prisma guide, which is provided by the journal for articles based on bibliographic reviews. This basically means considering including a Prisma flow diagram, selecting the diagram model considered most appropriate from the example models provided by the journal for this purpose. 

This diagram can also be completed using the items in the Prisma guide that the authors consider appropriate, in order to describe the review process and the selection of the bibliography used in the article.

Results: with regard to the indicators used, which come from quantitative studies performed through surveys, it could be useful to include a brief mention of the type of survey on which the document is based, the sample size, etc., wherever possible and where data is available.

Conclusions: according to the guidelines for authors, this section is not mandatory. However, including a brief section to summarise the main results and ideas from the article could be useful and provide clarification.

Bibliographic references: please check that the link included in reference 26 is correct.

Author Response

Dear Reviewer,
Thanks for your suggestions for improving this paper. Thus, from the 139-142 line we include the main aim in this paper. Further, thank you for your proposal for the Prisma diagram. Unfortunately, regarding that we have all published sources about sports innovations for Montenegro, we didn’t have any exclusions from applied content analysis. The potential main contribution of this paper would be the current development status of sports innovations in Montenegro, and the initial point for future research. So, we find it interesting to add Figure 2 for a clear understanding of collected published sources.
According to your proposal related to sample size, please find added two columns in Table 1 with information about sample size and applied instrument.
Checking reference 26 (in the corrected version reference 29 of the paper is done). Mentioned link works properly.

Please see the attachment. All corrections are colored in red.

Reviewer 2 Report

A small problem is methodology ref 26 cannot be found on the link.

The big problem: a review of the literature over the last 10 years (Google Scholar, PubMed, Scopus, and Web of Science) gives us just three papers from last year.
A review of governmental and non-governmental documents found only two documents.
Insufficient data sources for serious research.

Author Response

Dear Reviewer,
Thank you for your suggestions. Firstly, based on our checking mentioned reference 26 (in the corrected version of paper reference 29 can be found on the link. We hope that this time won’t be a problem with loading.
Regarding the second problem, we tried in this paper to prepare an important research paper based on all published references for sports innovations in Montenegro. We know that lacking many research papers is the weakness of this paper. On another side, we concluded that this topic in this country is not on a high level. So, this paper can contribute a lot as an initial and comparative point for future research on this topic.

Please see the attachment. All corrections are colored in red.

Reviewer 3 Report

While I think the method is appropriate I would have liked to see a larger pool of researchers score the studies to eliminate any specific bias that may have been connected to the two researchers. 

The authors provide some strong points in the discussion which may provide other countries with a model for sports innovation improvement. I do believe their methodological approach could use another round of review by another research to triangulate their results. I would also like to see the intercoder agreement reported. 

Author Response

Dear Reviewer,
Thank you for your opinion that we used the appropriate method and for all your suggestions. We included all published sources, evaluate them based on all recommendations for preparing the content analysis. We know that more researchers would provide better elimination of any specific bias. On another side, this is a new topic in this country and we did our best to set this initial point for further research on this topic.
All information about the inter coding agreement is reported from 162-169 lines and from 181-187.

Please see the attachment. All corrections are colored in red.

Reviewer 4 Report

This systematic literature review aims to review the literature on innovation in the field of sport in Montenegro. The results showed a low level of innovation in the field of sports in Montenegro was determined. Congratulations to the authors for the topic under study. However, this review study presents some methodological issues that may need to be resolved. In the comments below, some considerations are set forth.

Although the title reveals the scope of this study, it is suggested to clearly identify that this is a literature review study. It is possible to verify the existence of a high variety of review study methods. In addition, the title is of vital importance in selecting and reading review reports. It is recommended to identify the report as to the type of review that was conducted, being consistent with the methodological procedures described.

The abstract is organized in a well-structured format. However, this section can also include other aspects in order for readers to establish a more complete understanding of the findings. In this sense, it is suggested to include information regarding the literature search strategy (e.g., specify databases, eligibility criteria, synthesis method used).

Overall, the introduction is well organized. A framework of the problem is presented as well as the definition of key concepts within the study area. However, it is suggested that the authors situate this literature review, clearly and objectively, within the scope of other previously published reviews on the issue under study. It is suggested to indicate the gaps in knowledge that are intended to be filled, in the context of other works within the same scope. Furthermore, it is suggested that some of the phrases described may be supported in previous literature, giving it better robustness (e.g., "Also, the variability of a sports product contributes to new differences concerning organizations in other industries, because sports organizations can find it somewhat more difficult to guarantee the quality of a sports product or service compared to other organizations that produce consumer goods or services").

In order to reduce publication bias associated with studies that require high quality and rigor, avoiding the duplication of knowledge, and because of the implications that their conclusions may lead to, the registration of literature reviews is an important requirement for their publication. Therefore, it is recommended that this review work be registered (for example through PROSPERO, International prospective register of systematic reviews). Without the review protocol, how can we be sure that decisions made during the research process are not arbitrary, or that the decision to include/exclude studies/data in a review is not made in the light of knowledge about the results of individual studies?

It would be very important if the authors could clarify what eligibility criteria were used for the inclusion or exclusion of the studies in this review.

The search strategy presented is too vague. It is suggested that at least one, complete search strategy used should be documented in detail, at least for one database, so that it can be replicated by other researchers. Searching from any database, even by experienced researchers, can be imperfect. Thus, it is suggested that complementary research be conducted, to those conducted, considering other research resources such as checking the reference lists of the selected studies and direct contact with the authors of the papers to obtain manuscripts or missing data.

It is also suggested to describe how the studies were selected. Were all the studies read in their entirety? Were they selected first by title and then by abstract? Clarify.

In order to minimize possible risks of bias in the evaluation of the overall innovations, it is suggested that, for example, two reviewers independently evaluated the results, calculating the inter-judge agreement index using the kappa coefficient.

It is suggested that potential methodological biases that pertain to the development of the literature review be described. In addition, it may also be necessary to highlight the review's strengths from a methodological point of view.

Considering that many readers will read the conclusions of the literature reviews directly, it would be appropriate that, although described and analyzed in the discussion, the conclusions be presented in an objective manner and that they be based only on the evidence analyzed in the study.

Author Response

Dear Reviewer,

Thank you for your support in considering this important topic. Also, we try to solve methodological issues which need to be addressed in your opinion.

Although the title reveals the scope of this study, it is suggested to clearly identify that this is a literature review study. It is possible to verify the existence of a high variety of review study methods. In addition, the title is of vital importance in selecting and reading review reports. It is recommended to identify the report as to the type of review that was conducted, being consistent with the methodological procedures described.

Thank you for the suggestion that we need a correction of the title. We corrected the title to emphasize content analysis.

The abstract is organized in a well-structured format. However, this section can also include other aspects in order for readers to establish a more complete understanding of the findings. In this sense, it is suggested to include information regarding the literature search strategy (e.g., specify databases, eligibility criteria, synthesis method used).

Thanks for this suggestion. We corrected the abstract with information about the research strategy (electronic databases, etc.).

Overall, the introduction is well organized. A framework of the problem is presented as well as the definition of key concepts within the study area. However, it is suggested that the authors situate this literature review, clearly and objectively, within the scope of other previously published reviews on the issue under study. It is suggested to indicate the gaps in knowledge that are intended to be filled, in the context of other works within the same scope. Furthermore, it is suggested that some of the phrases described may be supported in previous literature, giving it better robustness (e.g., "Also, the variability of a sports product contributes to new differences concerning organizations in other industries, because sports organizations can find it somewhat more difficult to guarantee the quality of a sports product or service compared to other organizations that produce consumer goods or services").

Dear Reviewer, Thank you for your comments.

The intention of this paper is to upgrade the description from 108 lines to 116 lines. In these sentences, the authors try to explanate the basic intentions of this paper.

Based on your recommendations, we quoted references about the variability of sports products - Dees, Walsh, McEvoy, McKelvey, Mullin, Hardy, & Sutton, 2022).

In order to reduce publication bias associated with studies that require high quality and rigor, avoiding the duplication of knowledge, and because of the implications that their conclusions may lead to, the registration of literature reviews is an important requirement for their publication. Therefore, it is recommended that this review work be registered (for example through PROSPERO, International prospective register of systematic reviews). Without the review protocol, how can we be sure that decisions made during the research process are not arbitrary, or that the decision to include/exclude studies/data in a review is not made in the light of knowledge about the results of individual studies?

Dear Reviewer,

Thank you for your comments. According to applied methodology, which was more emphasized to content analysis than a systematic review, we didn’t realize registration in some of the mentioned regulations. As a final result, all authors have an attitude that we should consider your proposal in similar future research.

It would be very important if the authors could clarify what eligibility criteria were used for the inclusion or exclusion of the studies in this review.

Dear Reviewer,

Thank you for your comments. Based on that we haven’t many references for analysis (only 18 sources) we analyzed all sources. All applied process is presented in added Figure 2.

The search strategy presented is too vague. It is suggested that at least one, complete search strategy used should be documented in detail, at least for one database, so that it can be replicated by other researchers. Searching from any database, even by experienced researchers, can be imperfect. Thus, it is suggested that complementary research be conducted, to those conducted, considering other research resources such as checking the reference lists of the selected studies and direct contact with the authors of the papers to obtain manuscripts or missing data.

Dear Reviewer, thank you for your comment.

We included all available sources for this content analysis. According to that, in this country, it is not developed topic, methodology, etc., we have no opportunity at this moment to improve our findings. We can expect in the future, that we can develop and improve this methodology and strengthen further study with some new references. Also, we believe that our paper will provide a better accent on these important issues and contribute to the expansion of this topic on many researchers and stakeholders.

It is also suggested to describe how the studies were selected. Were all the studies read in their entirety? Were they selected first by title and then by abstract? Clarify.

Dear Reviewer, thank you for your comment. All published studies are included. When we add the name of the country „Montenegro“ in our process of searching, much research was excluded. All studies are completely read, including a title in the first place, and an abstract after that.

To minimize possible risks of bias in the evaluation of the overall innovations, it is suggested that, for example, two reviewers independently evaluated the results, calculating the inter-judge agreement index using the kappa coefficient.

Dear Reviewer, thank you for this great proposal. Based on that we describe all processes of evaluation of reviewers with the calculation of mentioned the kappa coefficient (lines 162-169 and 181-187). 

It is suggested that potential methodological biases that pertain to the development of the literature review be described. In addition, it may also be necessary to highlight the review's strengths from a methodological point of view.

Dear Reviewer, thank you for this valuable comment. Regarding that potential biases, operationalization was realized with key suggestions for content analysis: 1) operationalization of the concept of innovation in sports in the protocol of the study, 2) training coders for implementation of the formulated concept of sports innovation, and 3) evaluation of the implementation over the reliability of coders. The potential problem of risks of bias is solved by calculating the inter-judge agreement index using the Cohen's kappa coefficient (κ) suggested by McHugh (2012). Recommended values of this coefficient were between 0.81-1.00 as almost perfect agreement. Strengths of applied methodology reveal bring some new perspective of viewing on concept sports innovations. Researchers and practitioners often use the concept of sports innovations with more simplicity. Therefore, this paper can bring new directions for the conceptualization of sports innovations. So, the applied methodology could be improved with new ideas, concepts, etc.

Considering that many readers will read the conclusions of the literature reviews directly, it would be appropriate that, although described and analyzed in the discussion, the conclusions be presented in an objective manner and that they be based only on the evidence analyzed in the study.

Dear Reviewer, 

thank you for this comment. We did strengthen the conclusion in an objective manner. Please find mentioned corrections in lines 455-459 and 472-480.

Please see the attachment. All corrections are colored in red.

Round 2

Reviewer 2 Report

I still think there should be more resources for this kind of paper but considerable effort was put into the rest of the paper.

Author Response

ANSWER: Dear Reviewer, we appreciate so much your contribution and we do believe this manuscript is improved from the technical side so much accordingly. We have followed your instructions from Review Report (Round 1) and we have been relooking up all the available literature. However, it is not easy to find more resources for paper, mostly due to the reason, it doesn't exist. I would like to remind you the research of sport innovation was neglected in Montenegro and this research has been done within a national project under the title “Montenegrin Platform for Innovation in Sport” that was funded by the Ministry of Science to improve this area. From this reason, the main purpose of this study was to analyze all relevant and available scientific sources to evaluate the current status of sports innovation intending to set an initial point for future analyses of progress. This study is a pioneer study in this area in Montenegro but we do know it is not collected a huge number of documents, just available literature. The project team members worked on this research more than two months and we can confirm there are no more scientific articles, secondary literature, and online content, as well as world-leading reports that anybody can find with three keywords: "sport", "innovation" and "Montenegro". Therefore, we did not have anything against your suggestion, even we do believe it is a shame there are no more studies from this field in Montenegro, but we do believe this is the reality at moment and it will be much better in the future - the negative results of this study will certainly initiate decision-makers, researchers and practitioners to improve this field. For the reason that we would not like to offend you, we would appreciate it if you give us specific documents we missed to include in our content analyses, mostly due to the reason we really want to satisfy all your suggestions from the reviewing process. We do believe your comments were based on some certain mistakes you recognized we have done? In conclusion, we would like to thank you very much for your endorsement or prospective suggestions for the missing document from Montenegro.

Reviewer 4 Report

The manuscript can still be improved from the methodological issues. It is suggested that the present review paper be registered in PROSPERO (PROSPERO (york.ac.uk)).

Author Response

NOTE [Review Report (Round 2)]: The manuscript can still be improved from the methodological issues. It is suggested that the present review paper be registered in PROSPERO (PROSPERO (york.ac.uk)).

ANSWER: Dear Reviewer, we appreciate so much your contribution and we do believe this manuscript is improved from the methodological side so much. We have followed all your instructions from Review Report (Round 1) and we have clarified our method much more precisely and offer the potential readers to better understand what the main purpose of our research was. Although this is a blind process we can recognize your experience and expertise in this area, so we are so happy that the editors have succeeded to reach such an excellent reviewer. However, the last methodological issue that we have not done, was caused by the objective reason. Namely, we have visited the website you suggested and learn how to register our study within PROSPERO; however, the main reason we did not register our study was the following text: "PROSPERO accepts registrations for systematic reviews, rapid reviews, and umbrella reviews. PROSPERO does not accept scoping reviews or literature scans. Sibling PROSPERO sites register systematic reviews of human studies and systematic reviews of animal studies." (Cited from the following page https://www.crd.york.ac.uk/prospero/). From this reason, our study is not systematic, rapid or umbrella review, we were not able to complete registration. Therefore, we did not have anything against your suggestion, even we do believe it should not be mandatory, but our study is content analysis (literature scan) and it is highlighted at the front page that this type of studies PROSPERO does not accept to register. For the reason that we would not like to offend you, we would appreciate it if you give us the exact instructions on how to register our study on the subject portal, mostly due to the reason we really want to satisfy all your suggestions from the reviewing process. Thank you very much for your endorsement or prospective instructions for the registration.